# Participants’ Perceptions of “C.H.A.M.P. Families”: A Parent-Focused Intervention Targeting Paediatric Overweight and Obesity

**DOI:** 10.3390/ijerph16122171

**Published:** 2019-06-19

**Authors:** Kristen C. Reilly, Daniel Briatico, Jennifer D. Irwin, Patricia Tucker, Erin S. Pearson, Shauna M. Burke

**Affiliations:** 1Faculty of Health Sciences, Western University, London, ON N6A 5B9, Canada; kreill2@uwo.ca (K.C.R.); dbriatic@uwo.ca (D.B.); jenirwin@uwo.ca (J.D.I.); ttucker2@uwo.ca (P.T.); 2Faculty of Health and Behavioural Sciences, Lakehead University, Thunder Bay, ON P7B 5E1, Canada; erin.pearson@lakeheadu.ca

**Keywords:** childhood obesity, overweight, paediatric, parents, focus group, program evaluation, intervention, community, qualitative

## Abstract

*Background*: Recently, our team implemented a 13-week group-based intervention for parents of children with obesity (“C.H.A.M.P. Families”). The primary objective of this study was to explore, qualitatively, parents’ perspectives of their experiences in and influence of C.H.A.M.P. Families, as well as their recommendations for future paediatric obesity treatment interventions. *Methods*: Twelve parents (seven mothers, five fathers/step-fathers) representing seven children (four girls, three boys) with obesity participated in one of two focus groups following the intervention. Focus groups were audio recorded and transcribed verbatim and data were analyzed using inductive thematic analysis. *Results*: Findings showed that parents perceived their participation in C.H.A.M.P. Families to be a positive experience. Participants highlighted several positive health-related outcomes for children, families, and parents. Parents also underscored the importance and positive impact of the group environment, specific educational content, and additional program components such as free child-minding. Recommendations for future interventions were also provided, including greater child involvement and more practical strategies. Finally, parents identified several barriers including socioenvironmental issues, time constraints, and parenting challenges. *Conclusions*: Researchers developing family-based childhood obesity interventions should consider the balance of parent and child involvement, as well as emphasize group dynamics strategies and positive family communication.

## 1. Introduction

Obesity is widely recognized as one of the most significant health problems affecting children in the 21st century [1,2,3]. The prevalence of both overweight and obesity among children has increased dramatically over the last 30 years, with recent estimates showing that nearly 340 million children are affected worldwide [1]. The consequences of childhood overweight and obesity are severe and concerning. Children with obesity are at an increased risk of experiencing co-morbidities including type 2 diabetes [4], insulin resistance [5], metabolic syndrome [6], high blood pressure [4,7,8], non-alcoholic fatty liver disease [9], and asthma [10]. Childhood obesity has also been associated with negative and serious psychosocial outcomes such as depression [11] and reduced quality of life [10,12]. Furthermore, children with overweight and obesity tend to carry excess weight into later life [13], which can lead to the development of additional health consequences during adulthood, including stroke [14], osteoarthritis [15], and some cancers [15].

Consequently, there is an urgent need to design and implement effective and sustainable interventions that target the treatment of paediatric overweight and obesity [16]. One type of intervention, widely accepted and utilized in the treatment of childhood obesity, is the family-based approach [17,18,19,20]. Family-based paediatric obesity interventions acknowledge the family environment as a unit, as well as the significant influence of parents both as gatekeepers and role models, on children’s health-related choices and behaviours [17,21,22,23,24,25]. Thus, these treatments typically focus on improving factors such as parental support, family dynamics, and aspects of the home environment to enhance health-related behaviours among children [17,19,20,25].

Central to the family-based approach, ‘parental involvement’ has been identified as a key component of successful paediatric weight management interventions [18,26,27]. Kitzmann and colleagues (2010), for example, conducted a meta-analysis containing 125 experimental childhood overweight/obesity treatment studies to examine the effectiveness of interventions with high parental involvement (i.e., parents participated in all components of treatment) versus those with low parental involvement (i.e., only the child participated in the majority of the treatment components). The results showed that overall, childhood overweight/obesity treatment interventions consisting of high levels of parental involvement were significantly more effective with regard to improving child weight-related outcomes (i.e., weight, body mass index (BMI), standardized BMI (BMI-*z*), and percentage overweight) than were interventions with low levels of parental involvement [26].

Given the relative success of paediatric obesity treatment programs involving parents, researchers have also implemented and evaluated interventions that target parents as the “primary agents of change” [27,28,29,30,31,32]. Parent-focused interventions, also known as “parent agent-of-change” [30,31,33] or “parent-only” [28,34,35,36,37] interventions, are those that exclusively target parents in the treatment of childhood overweight/obesity [33,35]. While the primary outcomes generally remain child-focused, children are not directly involved in the intervention. Parent-only childhood obesity interventions have taken various forms based on focus (e.g., positive parenting skills [30,34], health knowledge/education and behaviour change [38,39], environmental modifications [40], etc.) and setting (e.g., primary care [41,42], out-patient [43,44,45], university [39], and community [46,47]).

Generally speaking, childhood obesity treatment studies in which parents have been identified and included as the primary agents of change have resulted in reductions in children’s BMI-*z* [28,45,48] and BMI percentile [41,42]. In addition, the authors of various systematic reviews have found that parent-only interventions are either as effective as [35,36,37] or potentially more effective than [37] family-focused (i.e., parent- and child-focused) interventions in terms of reductions in children’s BMI-*z* scores. Lastly, there is also evidence to suggest that parent-only childhood obesity interventions may be more cost-effective than traditional family-based interventions, as they are typically less expensive to implement and require fewer resources [37,49].

Approximately a decade ago, our research team developed and implemented a 4-week, family-based (i.e., parent-child) childhood obesity intervention, entitled the Children’s Health and Activity Modification Program (i.e., the original “C.H.A.M.P.” program [50]). This 4-week group-based pilot program was delivered to 40 families over two consecutive years in the form of a summer day-camp for children (Monday–Friday, 9:00 a.m.–4:00 p.m.) plus weekend education/activity-based sessions for parents (Saturdays from 10:00 a.m.–2:00 p.m.). Overall, qualitative data indicated that C.H.A.M.P. was received positively by both children [51] and parents [52]. The quantitative results were also promising; significant improvements were found for children’s fat and muscle mass percentages from pre- to post-intervention, and significant reductions in BMI-*z* were sustained 6 months post-intervention [53]. Perhaps most noteworthy were the significant improvements in child- and parent-proxy reported quality of life, sustained up to 12-months post-intervention [53], as well as improved physical activity self-efficacy from pre- to post-intervention [54].

Interestingly, qualitative data gathered via focus groups conducted after the original C.H.A.M.P. program also revealed that parents and children expressed a desire for greater parental involvement [51,52]. For example, many children expressed that they required more support and participation from their parents in helping them to adopt and maintain healthy behaviour changes [51]. In addition, C.H.A.M.P. parents noted that they would have liked additional education and program engagement opportunities (e.g., professional consultation, take-home materials, hands-on learning activities) and that they wanted program staff to hold them more accountable for lifestyle changes and participation in the program [52]. However, despite expressing a need for more involvement in the program, parental adherence to the C.H.A.M.P. intervention was low in comparison to that of children (i.e., 69% vs. 91% over 4 weeks). Indeed, such findings are in line with the literature as participant adherence and attrition issues have been cited as important barriers to and limitations of other childhood obesity interventions, particularly those that target parents [35,37,55].

On the basis of evidence from the original C.H.A.M.P. program (including parents’ perceptions of the program and recommendations for future interventions [52]), as well as the growing literature and documented effectiveness of childhood obesity interventions targeting parents, our team recently developed and implemented a 13-week group-based intervention entitled “C.H.A.M.P. Families”. Whereas the original C.H.A.M.P. program was offered primarily to children (with a relatively small family-based/parental component), C.H.A.M.P. Families was offered to parents (with minimal direct child involvement) who had a child with overweight or obesity. The overall purpose of the C.H.A.M.P. Families program of research was to implement and assess the feasibility of the pilot intervention using the RE-AIM (reach, effectiveness, adoption, implementation, maintenance) framework, a tool applied to facilitate the design and evaluation of health behaviour interventions [56,57,58,59]. Gathering information about participants’ perceptions of and experiences in such programs is a critical component of assessing intervention acceptability and feasibility [60]. Thus, the primary objective of the current study was to explore the perspectives of parents who participated in the C.H.A.M.P. Families intervention with regard to their experiences in the program, as well as the program’s influence on various aspects of child and parental wellbeing (i.e., health behaviours, parental confidence for supporting health behaviour change, and family communication). A secondary purpose was to explore parents’ perceptions of the program’s strengths and weaknesses, and to identify practical issues that could help to inform the design of future childhood obesity treatment programs. While previous studies have highlighted parents’ perspectives of their experiences related to primary care [61,62] and family-based childhood obesity interventions [63,64,65], to our knowledge, this is the first study to explore the perceptions of parents in the context of a community-based, parent-only lifestyle intervention targeting childhood obesity.

## 2. Materials and Methods

### 2.1. Intervention Description

C.H.A.M.P. Families was designed as a single-centre, single group, non-randomized prospective study. Grounded in a theoretical model integrating Social Cognitive Theory [66,67], group dynamics [50,68,69], and motivational interviewing techniques [70,71], C.H.A.M.P. Families was a 13-week intervention consisting of eight 90-min education sessions delivered to parents/guardians only (purposefully scheduled weekly and then bi-weekly to avoid an overreliance on the group [72,73]), as well as two post-program ‘booster sessions’ offered to guardians and children (see Reilly et al., 2018, for a detailed description of the study protocol and theoretical foundation [74]). All educational sessions were offered on Monday evenings at a local YMCA and covered a range of relevant child and family health topics including, but not limited to: child growth and development, nutrition, physical activity, sleep, sedentary behaviours, parenting and family dynamics, and mental health. Several experts (i.e., health professionals, researchers) and individuals from community organizations were invited to deliver intervention content to parents in interactive, group-based sessions. At the end of each session, parents received take-home materials, and were assigned “homework” activities to reinforce concepts and to assist parents in implementing lifestyle modifications with children in the home environment. Participation in C.H.A.M.P. Families was free and YMCA drop-in programming/child-minding was also available for all children, free of charge, while parents attended the educational sessions. All components of the study were approved by the host University’s Research Ethics Board and registered retrospectively with International Standard Randomised Controlled Trials Number (ISRCTN; ID# 10752416).

### 2.2. Participant Recruitment and Eligibility

Participants were recruited using a variety of strategies including newspaper and radio advertisements, social media, posters displayed in various community settings (e.g., libraries, local businesses, family health clinics), and study pamphlets and posters delivered to community paediatricians and family physicians. Parents/guardians were eligible to participate if: (a) they had a child between the ages of 6 and 14 years with a BMI ≥85th percentile for their age and sex; and (b) both the parent and child were fluent in English. All parents and guardians, including those living in separate homes, were invited to attend the program and participate in the study if interested and eligible.

### 2.3. Focus Groups

Two focus groups for parents/guardians and one for children were held concurrently during the last session of the formal intervention (i.e., in December 2017). Following a short end-of-program celebration during which the Program Coordinator (K.C.R.) and Principal Investigator (S.M.B.) presented families with participation certificates and awards, as well as a light dinner and refreshments, parents/guardians were asked to relocate to one of two focus group rooms within the YMCA facility (children remained in the main program location for their focus group; results from this focus group will be reported elsewhere). Parents were assigned by the Project Coordinator (K.C.R.) to one of two focus groups to ensure similar numbers in both groups. All focus group participants provided consent to participate, and in cases where more than one parent and/or guardian participated in the intervention, both were invited to participate in the same focus group to facilitate conversation and comfort.

Focus groups for parents were approximately 75 min in duration, audio-recorded, and transcribed verbatim. The Principal Investigator (S.M.B.) and a Masters-level graduate student (D.B.) moderated focus groups using a semi-structured interview guide developed based on relevant RE-AIM dimensions [56,57,58,59] and criteria for evaluating feasibility studies [60]. Once in their designated rooms, participants were reminded of the focus group’s purpose and procedures, and that participation was voluntary. To reduce the potential for socially desirable responses, parents were also told that there were no correct or incorrect answers, that their honest views and experiences were being sought, and were asked to keep the discussion confidential [75]. To begin the discussion, participants were asked to comment on their overall experience in C.H.A.M.P. Families (e.g., “How did you feel about C.H.A.M.P. Families and your family’s participation in the program?”). Next, participants were asked about their perceptions and the potential influence of C.H.A.M.P. Families with regard to: (1) child and parent physical activity and dietary behaviours (e.g., “What is different, if anything, for you about your own eating behaviours [or your thoughts about food and nutrition] since you started the C.H.A.M.P. Families program?”); (2) parenting confidence (e.g., “Do you find that you have higher levels of confidence in your ability to facilitate and support healthy choices in your family? If so, in what way(s)?”); (3) family communication and cohesion (e.g., “In what ways, if any, has your family’s communication changed since starting this program?”); and (4) barriers and facilitators to health behaviour changes (e.g., “Can you identify any barriers that might have impacted your child’s physical activity levels throughout this program?”). Finally, participants were asked to identify logistical issues (e.g., “How did you feel about the time commitment for this study?”) as well as considerations and recommendations for future programming (e.g., “How could the program be improved?”).

### 2.4. Data Analysis

Two researchers (K.C.R. and D.B.) reviewed the transcripts for accuracy and anonymized excerpts. The researchers subsequently analyzed the data in NVivo (Version 11.4, 2016; QSR International Pty Ltd, Doncaster, Australia) using an inductive approach in accordance with the six phases of thematic analysis described by Braun and Clarke [76,77]: (1) familiarising yourself with the data (i.e., multiple readings of the transcripts, noting preliminary ideas); (2) generating initial codes (i.e., systematically aggregating the data set into codes); (3) searching for themes (i.e., grouping codes and all relevant data into potential themes); (4) reviewing themes (i.e., creating thematic maps and confirming that themes accurately represent codes and data); (5) defining and naming themes (i.e., creating and refining names and definitions of themes); and (6) producing the report (i.e., choosing illustrative excerpts to exemplify data and connect it to the analysis, research question, and literature). The researchers conducted their initial analyses independently, and met subsequently with a third investigator (S.M.B.) to discuss and corroborate their findings. When discrepancies arose, the researchers discussed their interpretations of the data until agreement was achieved. Once a final consensus was reached for each theme, the researchers worked collaboratively to select a number of illustrative quotes. Several measures proposed by Lincoln and Guba [78] and adapted by Irwin and colleagues [79], including member checking, summarizing, and peer debriefing, were taken to ensure trustworthiness (i.e., credibility, dependability, confirmability, and transferability [78,80]) of the data and analysis.

## 3. Results

Twelve of the 16 parents (75%; 7 mothers, 5 fathers/step-fathers) who were enrolled in C.H.A.M.P. Families participated in one of the two focus groups (*n* = 6 participants per group). Participants (*n* = 12; *M*_age_ = 41.5, *SD* = 5.2; mean researcher-assessed parent BMI at baseline = 34.3 kg/m^2^, *SD* = 11.7) were parents/guardians of 7 children (*M*_age_ = 9, *SD* = 0.82; 4 girls, 3 boys; mean researcher-assessed child BMI-*z* at baseline = 2.20, *SD* = 0.28). Unfortunately, baseline BMI data were not available for one focus group participant, and age was not recorded for two participants. Parents who participated in a focus group attended an average of 85% of the educational sessions, a mean that was greater than that calculated for all parents in the program (*n* = 16; 73%). Additional demographic information for parents/families who took part in the focus groups is presented in Table 1.

The qualitative analysis revealed a total of 14 overarching themes and 28 subthemes. The following five categories containing themes and/or subthemes are described in detail below: (1) Outcomes for children (3 themes); (2) Outcomes for parents and families (3 themes, 8 sub-themes); (3) Impactful components of C.H.A.M.P. Families (3 themes, 7 sub-themes); (4) Barriers to health behaviour change (3 themes, 8 sub-themes); and (5) Recommendations for future paediatric overweight/obesity interventions (2 themes, 5 sub-themes).

### 3.1. Outcomes for Children

Parents noted that they had observed several positive changes in their children since beginning C.H.A.M.P Families. These outcomes were grouped into three overarching themes: improved dietary behaviours; increased physical activity; and enhanced empowerment and autonomy. Quotations exemplifying these three themes are displayed in Table 2.

#### 3.1.1. Improved Dietary Behaviours

Many parents expressed that their child(ren) displayed a greater awareness of their dietary behaviours and were making conscious efforts to choose healthier foods. Specifically, some parents described how their child(ren) showed more interest in the nutritional content of foods and were increasingly involved in meal planning and preparation throughout the duration of the program.

#### 3.1.2. Increased Physical Activity

A number of parents also noted that some children displayed greater motivation for and participation in various (and sometimes new) physical activities. Some parents mentioned that their child(ren) independently sought out opportunities to be active, and in some cases, attempted a new sport.

#### 3.1.3. Enhanced Empowerment and Autonomy

Parents discussed that their child(ren) seemed to feel more empowered and demonstrated greater autonomy over their health behaviours since starting the program. Many detailed how children appeared to be exhibiting greater control over their diet by preparing and/or choosing their own meals, as well as independently selecting and engaging in new physical activities.

### 3.2. Outcomes for Parents and Families

Parents were also asked about any personal, parenting, and/or family-related changes experienced as a result of their participation in C.H.A.M.P. Families. Three overarching themes and eight subthemes emerged within this category, including: healthy food choices for the family (i.e., healthier food purchases and food preparation at home); enhanced family dynamics (i.e., greater confidence to have conversations with children about weight, increased family communication, and full family engagement in health behaviour changes); and greater parental confidence to promote health behaviours in children (i.e., confidence to serve as the primary “agent-of-change”, enhancing children’s responsibility for their health [“letting go”], and perseverance towards change). Illustrative quotes for these themes and subthemes are presented in Table 3.

#### 3.2.1. Healthy Food Choices for the Family

This theme captured improvements experienced by several parents with regard to food purchasing and food preparation behaviours. For example, many parents reported that they were selecting healthier options at the grocery store or avoiding packaged foods or treats while shopping. A number of parents also described how at-home food preparation had changed for them as a result of the program, with many opting to prepare meals from scratch and/or substituting more nutritious ingredients for the less healthy products used previously.

#### 3.2.2. Enhanced Family Dynamics

This theme focused on parents’ perceptions of increasingly positive interactions and activities occurring within the family since beginning C.H.A.M.P. Families. Several parents noted that family communication had improved. More specifically, many parents noted that that they felt more comfortable in broaching the topic of weight with their children, and that these discussions had become easier to have as a family unit. Rather than focusing solely on the child, parents discussed the involvement of the entire family in working toward positive and sustained health behaviour changes.

#### 3.2.3. Greater Parental Confidence to Promote Health Behaviours in Children

Several parents expressed that they had experienced an increased level of confidence in relation to their knowledge and role as the primary ‘agent of change’ within the family unit and in the home environment. In addition to this enhanced confidence for supporting behaviour change, parents also reported that they were more confident in their ability to empower and provide their children with additional responsibility and control over their own health. Specifically, parents commented on the benefits of ‘letting go’ from a health perspective; described by many as acknowledging children’s abilities and preferences by allowing them to choose their own foods and assist with meal preparation. Finally, parents voiced that the despite the challenges and barriers experienced in promoting and changing health behaviours, they had the confidence to persist with the efforts required to achieve their family health goals.

### 3.3. Impactful Components of C.H.A.M.P. Families

The following three themes and seven subthemes were identified on the basis of parents’ responses regarding their perceptions of the effective components of C.H.A.M.P. Families: group environment (i.e., sense of community/belonging and group interaction and support); program experts and information (i.e., inspiring and motivational expert speakers, relevant and applicable information and resources, reminders and reinforcements); and additional program benefits (i.e., complimentary programming for children and at-home data collection visits, tools, and personnel). Quotes exemplifying these themes and subthemes are displayed in Table 4.

#### 3.3.1. Group Environment

The positive aspects and significance of the group environment were emphasized often by parents, particularly in reference to the sense of community and belonging that were generated. Parents expressed the importance and benefits associated with feeling as though they were part of a group, and that they were not alone in their struggles to improve their child’s health. Many parents also noted that the group-based focus of the program, as well as the social support provided by program personnel and other participants, were very impactful. Parents spoke pointedly to the power of hearing other families’ experiences and struggles, and the significant impact of group problem solving.

#### 3.3.2. Program Experts and Information

Many parents stated that the information delivered throughout the program was both relevant and applicable to their child and family, and they highlighted specific ways they were able to use the resources received (e.g., C.H.A.M.P. Families binder, children’s kitchen utensils, posters and readings). In addition to the usefulness of program information and resources, parents noted that the intervention agents (i.e., the “experts”) delivering the content were also highly impactful. Several participants noted that they found the guest speakers to be inspiring and motivational, highlighting the positive impact of the sessions delivered by the professional chef, the dietitian, and the public health nurse who specialized in mental health specifically. Finally, parents expressed that the program content and materials bolstered parents’ existing knowledge about health and that the program itself served as a nudge or a reminder to prompt behaviour change.

#### 3.3.3. Additional Program Benefits

Parents commented on a number of additional program components that were not part of the formal intervention delivered to parents (i.e., components that took place outside of the group-based sessions). For example, several parents stated that the complimentary YMCA programming offered to children through the C.H.A.M.P. Families program was perceived very positively by both themselves and their children. Having a structured, easily accessible, and safe activity program available for children to engage in—with other children whose parents were enrolled in the program—was identified as being very important to and an unexpected benefit of the program for most parents. Furthermore, parents commented positively on the research-related components of the program, namely the home data collection visits and the positive relationships developed with program personnel. Specifically, parents described how home visits with the Project Coordinator were important for establishing trust, ensuring comfort, and facilitating dialogue within the family unit and among parents, children, and C.H.A.M.P. staff. Parents also expressed that the research tools used and administered/distributed during home visits (i.e., questionnaires and accelerometers) were a source of motivation for children.

### 3.4. Barriers to Health Behaviour Change

Insofar as barriers and challenges related to changing health behaviours are concerned, the following three overarching themes and eight subthemes were identified based on parents’ responses: socioenvironmental issues (i.e., school-related issues, stigma and bullying, social pressures and the food environment, lack of flexible and cost-effective programming for children, and geographic and seasonal issues); time constraints; and parenting issues (i.e., protecting children’s feelings, setting appropriate boundaries, and difficulties associated with relaying program content to children). Illustrative quotes for these themes and subthemes are found in Table 5.

#### 3.4.1. Socioenvironmental Barriers

Parents identified a broad range of socioenvironmental barriers that affect both themselves and their families. First, some parents reported that they experienced feelings of shame and perceived disapproval from others for having a child with overweight or obesity. Furthermore, they spoke of the bullying and discrimination some of their children experienced as a result of their weight. Second, parents noted specific issues related to the school system, including children’s perceptions that school lunch and snack breaks were too short. Many parents felt that the lack of time available for children to eat during school hours had a negative impact on their diet in that children would either have to rush to eat at an unhealthy pace or leave food uneaten. Parents also voiced concern about the perceived lack of health, nutrition, and physical activity-related education their children were receiving at school. The third subtheme, social pressures and the food environment, captured parents’ perceptions of the social forces (i.e., family, peers, cultural, and societal norms) as well as the physical presence of and proximity to food that promote the consumption of unhealthy foods. Several parents spoke about the challenges of maintaining a healthy diet during holidays, at family functions, and during parties when treats and other unhealthy foods are readily available. One parent spoke to the pervasiveness of food marketing and how it affects children’s food preferences and attitudes. The fourth theme pertained to parental perceptions of the lack of flexible and cost-effective programs available for children in the community, referring most often to a lack of informal, inexpensive, physical activity programming for children. Many parents felt that current programs were overly structured and competition-focused which was discouraging for some children and required families to commit for several weeks/months (often without a trial period). The fifth and final subtheme, geographic and seasonal issues, referred specifically to the barriers to health behaviour change that parents identified (e.g., cold winter months, living in a small community with few resources) in relation to weather and location.

#### 3.4.2. Time Constraints

The perceived lack of time, and/or inability to manage time effectively, to prepare healthy meals and engage in physical activity was identified by parents as a significant barrier to sustained health behaviour change. Many parents suggested that between work, school, and extra-curricular activities and responsibilities, it was challenging for both parents and children to schedule time for grocery shopping, meal preparation, and physical activity.

#### 3.4.3. Parenting Issues

The following three subthemes related broadly to parenting were identified as barriers to healthy behaviour change: protecting children’s feelings, setting appropriate boundaries, and difficulties relaying program content to children. A number of parents voiced concerns about hurting their children’s feelings, damaging their self-esteem, or unintentionally creating other issues by discussing weight- and health-related topics with them. Parents also expressed that they found setting appropriate limits related to food and screen time challenging, as many felt they were being overly withholding or restrictive to children. Finally, several participants noted that because their children did not view them as an ‘expert’ or authority on health, parents’ ability to relay and share the knowledge and information gained during the intervention with children at home was difficult and not always well-received. Some parents suggested that children would be more open to receiving this information, and therefore more likely to change their behaviours, if the intervention were delivered to the children themselves by experts.

### 3.5. Recommendations for Future Paediatric Overweight/Obesity Interventions

Two overarching themes and five subthemes resulting from parents’ recommendations for future interventions were generated, including: greater child involvement (i.e., increased accountability of children, hands-on activities for children, and peer supports and interactions among children) and practical information and strategies (i.e., missing the ‘how’ to follow through on lessons learned and other topics of interest). Quotes reflecting the abovementioned themes and subthemes are presented in Table 6.

#### 3.5.1. Greater Child Involvement

Most parents voiced a preference for children to be more directly involved in the intervention. Specifically, parents expressed that by adding a structured, child-focused program component, children would be more empowered, as well as more conscientious of, committed to, and accountable for their health behaviours. Parents provided a number of suggestions for future programs, including the inclusion of opportunities for children to participate in practical learning (i.e., ‘hands-on’) learning experiences such as cooking classes. Parents also spoke to the importance of and benefits associated with fostering additional peer support and interactions among children. Many noted that engaging children in group activities would have helped to develop additional friendships and supportive relationships among children who experience the same issues and challenges. Parents felt that the group-based format had been a very impactful component of the program for themselves, and that creating a similar environment for children would have had a powerful and positive effect on their children as well.

#### 3.5.2. Practical Information and Strategies

The final theme is divided into two subthemes that focus on information (i.e., other topics of interest) and strategies (i.e., how to implement the information learned during C.H.A.M.P. Families). Parents expressed an interest in different topics related broadly to child health such as weight-related communication and emotional intelligence. Parents also stated that time management strategies would have been helpful, as time constraints were identified as a hindrance to healthy eating and physical activity in their family. Many parents voiced that while they had sufficient information on certain topics, they still felt they lacked concrete strategies pertaining to how to implement this knowledge with their children and families in the home environment.

## 4. Discussion

The purpose of this study was to explore parents’ perspectives related to their role(s) as the ‘primary agent-of-change’ in a parent-focused childhood overweight/obesity program, as well as the perceived impact of the program on child and parental health and wellbeing. Program strengths and weaknesses, as well as practical issues and recommendations that could contribute to the design of future family-based treatment programs for paediatric obesity were also elicited. Several studies have highlighted parents’ perspectives of their experiences related to primary care [61,62] and family-based interventions [63,64,65], but to our knowledge, this is the first study to explore the perceptions of parents in the context of a community-based, parent-only lifestyle intervention targeting childhood obesity.

The parents who participated in focus groups described several perceived benefits for children (i.e., improved dietary behaviours, increased physical activity, and enhanced empowerment and autonomy), families (i.e., enhanced family dynamics and healthy food choices), and themselves (i.e., greater parental confidence to support and promote health behaviours in children), all of which were attributed to their involvement in the program. One additional and unanticipated benefit of the program that was highlighted by many parents related to the free, active programming that was offered to children at the YMCA during the parent-only sessions. Though this programming was not part of the formal intervention and was originally intended to reduce barriers to participation, it was noted by participants to have very positive outcomes for both children and parents. Interestingly, while parents noted improvements in their confidence to serve as agents of change for their families and to have conversations with children about health- and weight-related issues, they also emphasized that these areas could be addressed more explicitly in future paediatric obesity treatment programs. For example, several parents articulated challenges associated with relaying program content to children, suggesting that while they felt they had sufficient knowledge about the health topics discussed during the sessions, they lacked the necessary tools and strategies to effectively implement changes in the home environment. Some parents also noted that their children would likely be more receptive to the information if it came from an “expert” rather than from a parent or guardian. With regard to communication, many parents expressed a desire to protect their children’s feelings and self-esteem, which they believed could be damaged if they did not broach certain health- and weight-related topics sensitively.

Indeed, poor family communication has been found to be associated with an increased risk of child overweight/obesity [81], and certain types of parent-child weight-related talk has also been identified as potentially detrimental to a child’s health and wellbeing [82]. For instance, in a 2016 meta-analysis consisting of 4 intervention studies and 38 associative (cross-sectional and prospective) studies, Gillison and colleagues found that communication consisting of weight criticism (i.e., teasing) and encouraging weight loss increases the likelihood of poor physical self-perceptions, dysfunctional eating, and dieting behaviours in children [82]. Conversely, Gillison et al. reported that encouraging healthy exercise and diet without discussing weight directly was associated with less unhealthy weight control and dieting behaviours among children [82]. Unfortunately, evidence-based resources and strategies to help parents navigate conversations with children about food and weight management are lacking in the literature [81,82]. Furthermore, it important that as researchers, we acknowledge the possibility that we may inadvertently draw parents’, and subsequently children’s, attention to weight given that weight-related measures such as BMI-*z* are often the primary outcome in childhood obesity studies [35]. Thus, shifting the focus towards healthy lifestyles and facilitating positive and supportive family communication are important considerations for future paediatric overweight/obesity interventions [82]. Additional barriers to health behaviour change identified by C.H.A.M.P. Families participants, including time constraints, parenting issues, and lack of social support, were consistent with those that have been previously cited by parents in the childhood obesity treatment literature [62,63,64].

As noted previously, C.H.A.M.P. Families was informed by feedback from parents who took part in the original C.H.A.M.P. program [50], many of whom advocated for greater parental involvement and accountability in future paediatric obesity interventions [52]. Despite evidence indicating that parent-only interventions for childhood overweight/obesity may be as effective, or even more effective, than parent-child interventions [37], many of the parents in the current study noted that their children would have benefited from increased participation in the program. Taken together, it is reasonable to suggest that parents seem to desire a childhood obesity treatment program that is relevant for, and balances the involvement and accountability of, both parents and children.

While nearly all of the feedback about C.H.A.M.P. Families was positive, one parent did note that the delivery of content provided by one of the invited guest speakers was not relatable or relevant to their family. Although this comment was not deemed to be sufficient to warrant its own theme per se, such feedback will certainly be used by our team in the development of future programs.

One of the most impactful components of C.H.A.M.P. Families identified by participants was the sense of community and belonging that developed among the parents in the program. Connecting with other parents in a group-based setting was perceived by parents, especially those who had experienced stigma associated with having a child with overweight/obesity, as very powerful; many noted that they valued feeling as though they were “not alone”. This finding stresses the importance of cultivating a positive and inclusive group-based environment to support health behaviour change [83]. Groups can be powerful facilitators of change for and adherence to a variety of health behaviours [83,84,85,86], and in the context of childhood obesity, group-based programs have been shown to be more effective in reducing child BMI-*z* scores than treatments administered individually [87,88]. As stated previously, C.H.A.M.P. Families was intentionally designed using several evidence-based group dynamics strategies [74] that have been used successfully in previous family-based childhood obesity interventions [50] in an attempt to enhance adherence, group cohesion, and other health-related outcomes.

In addition to the importance of the group environment, participants emphasized that their experience in the program was enhanced by the rapport developed between themselves (including their children) and the Project Coordinator whom they described as likeable, engaging, and non-judgmental. Weight bias among primary care providers [89,90], as well as exercise and nutrition professionals [91], has been well-documented in the literature and has been shown to compromise patient outcomes and quality of care [92]. Furthermore, perceptions of judgment from health professionals can have a negative effect on weight loss [93]. While this intervention was administered by researchers in a community setting, it remained important for program staff to foster supportive relationships with participants to ensure that they felt accepted and did not experience stigma or judgment.

The present study is not without its limitations. First, given that the focus groups were moderated by members of the research team, it is possible that the positive feedback received from participants was influenced by social desirability [94], despite the use of honesty demands [75]. Second, the focus groups were conducted immediately following the intervention which might have increased positive perceptions related to the program and also limits the researchers’ ability to capture participants’ long-term perspectives of the program. Third, while the majority of C.H.A.M.P. Families participants (75%) attended a focus group session, there were four participants who did not participate in the focus groups (one who withdrew from study and three who had scheduling conflicts) and thus, whose perspectives and experiences were not captured. Three of the four participants who did not partake in the focus groups reported lower than average household income and/or education levels, and the same number of participants (although not necessarily the same individuals) attended ≤ 50% of the C.H.A.M.P. Families sessions. Given our small sample, it is unclear whether these factors impacted program participation or effectiveness, or whether the current findings might have differed had these individuals shared their experiences in a focus group. Fourth, it should be noted that despite efforts to recruit a diverse sample of participants, the individuals who participated in these focus groups, and in C.H.A.M.P. Families overall, were fairly homogenous in terms of their ethnicity and socioeconomic factors (i.e., household income and education). This is in line with a previously noted limitation in the childhood obesity intervention literature in which individuals of ethnic minorities and low socioeconomic status tend to be under-represented [55]. Further research examining strategies to improve recruitment, engagement, and adherence of these individuals is warranted. Lastly, as a result of the limited sample size, it was difficult to assert with confidence that true data saturation was reached.

## 5. Conclusions

Given the current prevalence of childhood obesity, there is an urgent need for treatment programs that are feasible, effective, and accessible to parents and families [27]. Based on participants’ perceptions, C.H.A.M.P. Families appears to have been well-received, and to have had an overall positive influence on the health and wellbeing of both parents and children. Further research exploring the development and dissemination of effective communication strategies related to weight and other sensitive health-related topics for families is necessary. Finally, group dynamics strategies should be used to enhance perceptions of belonging among families, and positive family communication should also be emphasized in future childhood overweight/obesity treatment interventions.

## Figures and Tables

**Table 1 ijerph-16-02171-t001:** Demographic information for parents who participated in the C.H.A.M.P. Families focus groups.

Demographic Variables	*n* (%)
*Gender*	
Female	7 (58.3)
Male	5 (41.7)
*Ethnicity*	
White/Caucasian	10 (83.3)
Arab	2 (16.7)
*Marital Status*	
Married	9 (75)
Common-law	2 (16.7)
Single, never married	1 (8.3)
*Level of Education (n = 10)*	
University degree (or higher)	6 (60)
College diploma	3 (30)
Post-secondary qualification	1 (10)
*Annual Household Income (n = 10)*	
$100,000 or more	6 (60)
$50,000–$99,999	3 (30)
$49,999 or less	1 (10)

Note: Two parents/guardians did not complete the full demographic questionnaire.

**Table 2 ijerph-16-02171-t002:** Selected quotes related to parents’ perceptions of outcomes for children.

**Improved Dietary Behaviours**
“As opposed to [packaged food], now they’re [my children] going for like the banana or the orange, something more healthy when going for that quick snack. I think they’re more aware of the calories too now.” (Participant 6, male) “The program has ignited that [initiative] in him [my son]. At home he is reading labels right now…he is counting calories…he is just little bit more aware…. Since the program he wanted to start helping out and prepare meals.” (Participant 11, female) “Not that we were thinking a pop a night or anything, but I would say the sugary drink consumption has gone down significantly in the household.” (Participant 1, female) “He’ll [my son] always want to talk about portion sizes now and reading the [nutrition labels] on products.” (Participant 5, female)
**Increased Physical Activity**
“[My son] started hockey this year, but not just that, even at school before he was just hanging out with his buddy, but now he’s playing soccer.” (Participant 10, male) “[My son] came home and said he was on the volleyball team, and he’s not the child that would ever sign up for anything.” (Participant 9, female) “He [my son] will set a timer and say, ‘I need to go to the park for an hour on the bike.’… He’s wanting to participate in things, wanted to use the treadmill.” (Participant 11, female)
**Enhanced Empowerment and Autonomy**
“One thing she’s [my daughter] taken upon herself is to make her own eggs for breakfast. She’ll have eggs with cheese and a little piece of fruit and water. Not that she didn’t eat eggs before, but she made them herself now.” (Participant 1, female) “I let [my daughter] make dinner, and giving up the control and saying OK you can do that. I didn’t think she could be able to it, but…she gave me, you know, the idea that she really was old enough to help us do more in that way.” (Participant 7, female) “Our daughter tried out for volleyball, which I’m not so sure if she would’ve done that before…she didn’t make the team but she was confident enough to go…. I think there is a healthier diet and she’s healthier and stronger so she’s willing to take on more and take those risks, and if you don’t make it, that’s OK.” (Participant 1, female)

**Table 3 ijerph-16-02171-t003:** Selected quotes related to parents’ perceptions of outcomes for parents and families.

**Healthy Food Choices for the Family**
i. Healthier Food Purchases “We go one night a week for our treat night, but before it was any day of the week.” “I would say the [biggest change is] packaged food. We’re not buying near as many chips anymore.” (Participant 5, female) “I’m particular about what I buy [the] kids…. I’m more aware of what I’m doing to give them a better chance of becoming healthy so, and staying healthy.” (Participant 9, female) “We still have some granola bars, but not nearly the amount of packaged cookies and things like that.” (Participant 1, female)
ii. Preparing Healthier Meals at Home “Instead of just whipping up a batch of chocolate chip cookies, because the kids love chocolate chip cookies, I’m looking for healthier choices in a cookie that I can hide. Now I make a quinoa chocolate chip cookie that the kids think is just a regular chocolate chip cookie, but it’s fortified with a whole bunch of stuff that they can’t see.” (Participant 9, female) “Something we took away from this was preparing meals ahead of time, like the day before for the next day so it would be a healthy meal and not a rushed out of the box meal sort of thing.” (Participant 1, female) “I think it made me concentrate more on what I was putting in front of my kids as food like…even if it was something simple or maybe it wasn’t the best things they can be eating, how can I just make this a little more appealing, healthy… like instead of box of craft dinner like make homemade not as much, use skim milk, not as much cheese, just try and make it a little healthier… it just made me think more about what I was actually feeding my kids.” (Participant 8, male)
**Enhanced Family Dynamics**
i. Increased Family Communication “… so as soon as supper’s ready…[everyone] participates in …. We talk amongst our supper table now and it’s much better” (Participant 5, female) “Every time we leave here [C.H.A.M.P. Families], almost the whole drive back home he’s [my son] talking ‘What’d you learn, Mom?’.” (Participant 5, female) “It’s an interesting conversation…unfolding every day at our dinner table. It’s fun to hear them [my children] …we’re so busy, you don’t get those little bits of pieces … you see all this coming out the dinner table, it’s a good thing.” (Participant 9, female) “It was a good opportunity for us to be more on the same page…. I think that’s a big impact to have both parents on board…and then be able to go back to the kids and say a five-minute synopsis of this is what I learned tonight.” (Participant 1, female)
ii. Greater Confidence to Engage in Health- and/or Weight-related Conversations with Children “I think for me it was not a license, but…it’s OK to talk to her [my daughter] about being overweight and about us as a family being overweight, and being healthier. It was almost like it made it feel like it was OK to have those conversations and feel more comfortable with the conversation... it was an elephant in the room amongst the family that was just gone” (Participant 1, female) “… you always want to protect your children, so sometimes you might see an issue or see something but you don’t wanna address it because you’re afraid of their reaction. So you go along your merry way and if you keep doing the same thing nothing changes. So if you bring everything out in the open and start to talk, then once communication opens up you get a lot of feedback and you learn what motivates your children too.” (Participant 10, male) “I think too for us this kind of explains to [my son] that there’s different types of body types. You might always be wider [or] heavier because you’re just built like that….” (Participant 6, male)
iii. Full Family Engagement in Health Behaviour Change “… the whole family’s doing it now, where before it was always let the kids go out and play.” (Participant 10, male) “As far as the whole family trying to participate in eating healthier and being more active, while trying to achieve the same goal and trying to help [my son] get where he wants to be and where we want him to be…. we are working together.” (Participant 6, male) “… you’re doing it as a whole family instead of an isolated individual.” (Participant 11, female) “Involving [our son], who is not overweight…. We said it was a family thing, whereas before it has always been [our daughter] do it and not [our son], right? So that’s a big thing the whole family was involved.” (Participant 1, female) “So we are always at the arena, all three of them [my children] play hockey. So if one of them is down there now all of us walk the walking track instead of sitting in the seats.” (Participant 5, female)
**Greater Parental Confidence to Promote Health Behaviours in Children**
i. Confidence to Serve as the Primary “Agent-of-Change” “I do feel as though I’m more confident. I can tell her [my daughter] why we’re doing what we’re doing now. I have more knowledge, so I can pass that to her.” (Participant 8, male) “I am a lot less likely to make an excuse as to why I can’t go out with her [my daughter]. If she wants to do a dance party in the basement or if she wants to whatever. I have to be there. I have to at least make it possible for her to do it.” (Participant 3, male)
ii. Enhancing Children’s Responsibility for Their Health (“Letting go”) “I never realized how big of a deal that letting go is, that’s huge. I wouldn’t know that if they [guest speakers] didn’t say that. Let them [the children] do it, that’s big for them.” (Participant 3, male) “I think I have learned some things that have helped my daughter, or given me the courage to do things a little differently, or let her try things like, let her cook one night and use the sharp knives that I wouldn’t have ever thought she was capable of doing. She is so much more capable than I ever gave her credit for.” (Participant 7, female) “Our kids are doing stuff in the kitchen. Even though it ends up taking longer to make the meal, they’re involved and I’m also making sure to protect that time.” (Participant 3, male) “I am very particular…and I like things to done a certain way, so knowing that it’s OK to let it go and let them be responsible, it’s a huge new thing for me.” (Participant 9, female)
iii. Perseverance Towards Change “You need to keep coming back, keep being reminded that in the end things could change if you try and keep trying.” (Participant 4, female) “Just trying to open up and be more mindful … I’m trying different things.” (Participant 1, female) “I found setting goals hard because you’re trying to set the goals to correct the difficult situation … but that is something that’s going to go forward and I’ll use the information we got to help with that.” (Participant 3, male) “It takes time … it’s hard for parents … there is nothing wrong with your kid or the way we do it, it just takes time.” (Participant 11, female)

**Table 4 ijerph-16-02171-t004:** Selected quotes related to parents’ perceptions of impactful components of the C.H.A.M.P. Families program.

**Group Environment**
i. Sense of Community/Belonging “It’s nice just meeting other families and knowing that we’re not the only family that’s having a child that is overweight…. Your circles of friends have children that maybe don’t have obesity or overweightness, and it’s hard to talk to them or you wouldn’t talk to them because you can’t relate. It’s nice to know that there are other people with the same kind of issues, that have struggles that are real like yours.” (Participant 1, female) “For me just listening to all the other parents and their struggles … that we’re not the only ones, that’s what I think was important for me.” (Participant 6, male) “The strength of the program is the community feeling that you generate…There were a lot of emotions that parents had…and in there [C.H.A.M.P. Families] they are kinda laid out. I think that actually brings the group closer.” (Participant 3, male) “We are not alone. Sometimes in the thick of it you think my god I’m like the only parent who has this problem? What are we doing wrong and why? How come everyone else has it figured out and I can’t get this figured out?” (Participant 4, female)
ii. Group Interaction and Support “I love the fact that it’s a support program for the families. I work in healthcare and I haven’t seen a program like this … It’s good to see that there is something going on that helps parents get together and learn from others … like this is an idea I can implement, this is something that we can do, this is something that we are missing in the community…. so, it has been really rewarding.” (Participant 11, female) “I like the program because it was a very positive environment and it was nice to know other parents’ concerns and how they approach situations, or just to know there are common issues that I didn’t know how to solve…. just to share the feelings and what worked and what doesn’t.” (Participant 3, male) “Listening to other parents I was like, OK, I can take from that feedback and apply it to myself to change how I deliver my message tonight on how he [my son] needs to care for himself, maybe allow him to do that.” (Participant 9, female)
**Content and Materials**
i. Inspiring and Motivational Expert Speakers “I really enjoyed the mental health speaker … I could relate better…with my kids, so I thought that was really useful.” (Participant 5, female) “As the person who does the majority of the grocery shopping, I would say that the grocery tour too has a big tangible impact … our dietitian that was with us was able to point out things … it has just changed the way that I shop, which changes the way that you prepare food and you kinda stay away from certain things and it has [an] impact on a whole household.” (Participant 1, female) “The biggest thing that I learned was when he [expert chef] said to have your children take control of the plate and their meal … that never dawned on me before. Having them [my children] serve themselves and put whatever on their plate, and putting some guidelines on there, but not saying you have to eat it all … that was profound to me.” (Participant 1, female)
ii. Relevant and Applicable Information and Resources “I’m gonna use that book [C.H.A.M.P. Families binder containing program resources and goal setting worksheets], keep referring to it. I’m going to fill in the parts that I didn’t do and use it as a guide. If I find something good, I’ll add to it. Gonna be kind of the reference.” (Participant 3, male) “We got a lot of good information as to how to go about doing it, I think the key is to apply it to your individual circumstances.” (Participant 3, male) “I would say the sugary drinks like, we had that little paper that we got that night on the fridge for a long time, and now we all know … that was a big eye-opener.” (Participant 1, female)
iii. Reminders and Reinforcements “There was a constant reminder. You want to slack off [then] you remember the program. You think back to it and say, ‘We are doing well, let’s get back to where we were.’ It was nice that it was a few times a month so even if you did forget, you could come in [on] the Monday, get the reminder again, go back home, refresh the information.” (Participant 12, female) “I think that [it] helped… because it takes so many tries. You need to keep coming back, keep being reminded that in the end things could change.” (Participant 4, female) “I think that for me coming here…. reinforced the fact that I am on the right track.” (Participant 9, female)
**Additional Program Benefits**
i. Complimentary Programming for Children “My favourite part was [for] my daughter. She had a blast. She absolutely loved it. And five times in the last week, she said: ‘I wished this wasn’t the last week of the C.H.A.M.P. Families. I wished we could go back again.’ She’s enjoying it so much. She loves the kids and she’s just had new experiences that we haven’t had before.” (Participant 7, female) “We are members of the Y, but we never come and that’s the thing, this brought me back….I used to go up to the gym and my kids. I would drop them off Monday and Thursday...and they used to love it…but with life, you just stop, but with a program like this you feel like you have to come in, you’re obligated, you signed up, you come in and it’s the same set of kids.” (Participant 12, female) “That connection with the kids that are similar to him, that’s what he [my son] really liked about it…. He is able to physically relate to this boy, he made friends with him. He hated the weeks we had no class. When I picked him up, he was like, ‘Mom, I was so happy all day and so excited looking forward to the program tonight.” (Participant 11, female)
ii. At-Home Data Collection Visits, Tools, and Personnel “The Actical thing [accelerometer] for [my son] was a very great competitive item. He knew that when he had it on he was focused and had to do whatever… and I know that it impacted the way that he looked at some of the things that he was doing on a regular basis.” (Participant 9, female) “He [my son] liked the small chats with [the Project Coordinator].” (Participant 11, female) “I found when [Project Coordinator] comes for the data collection, you don’t feel judged when you step yourself on the scale and your kids get on. There is no judgement and I think that is genuine.” (Participant 1, female) “I will say she [Project Coordinator] is special…. she just has this great way about her and the data collection. I’m filling out the survey and I’m doing my own thing, she’s chatting with [my son], and then she’s asking me the questions and they are having a little chit chat.” (Participant 9, female)

**Table 5 ijerph-16-02171-t005:** Selected quotes related to parents’ perceptions of barriers to health behaviour changes.

**Socioenvironmental Barriers**
i. School-Related Issues “If you look at this group…and the overall challenge with children today, eating disorders and being overweight, there just seems to be a real lack of focus on it in the school system.” (Participant 10, male) “Sometimes they’ll come back [and] they haven’t touched their vegetables or anything in their lunch, and when we challenge them on it it’s like we didn’t have enough time.” (Participant 2, male) “I think they might have gym once or twice a week [at school], and with no actual focus on what to eat, how to eat.” (Participant 2, male)
ii. Social Pressures and the Food Environment “You have to make the best decisions when you’re around the table and there’s a buffet, and Grandma made something, Great Aunt made something … so have a typical strategy that works best for you, cause you don’t want to feel that now you have limited yourself or restricted yourself, and now that becomes another forbidden fruit for you and for the family, it’s hard.” (Participant 11, female) “With my kids, no sugary drinks. When they go to parties and everyone is having juice or pop it’s a, ‘Why can’t we have carbonated beverages, Mom?’, or like when the nephews and nieces come over and they have pop for breakfast.” (Participant 11, female) “We are really competing against food science and some of the brightest and best food scientists out there you know. It does not take fifteen times for a snack food to appeal to a kid. It is like an addictive drug, it really is … how do you compete against that?” (Participant 3, male)
iii. Lack of Flexible and Cost-Effective Programming for Children “… one of [the] things is cost for a lot of families. If you’ve already got your kids in a lot of activities and then your kids want to go out and do things. Everything costs money, right?” (Participant 9, female) “Everything is so structured and monetarized now too, it’s a struggle. People think they need to go to these places to go do physical activity. It is a mindset and it doesn’t have to be that way.” (Participant 10, male) “I wanted a program like a nice supportive activity where I could drop her off once a week…. Where they do nice, structured, physical activities with a lot of other kids… Like a recreational but hard physical activity. Something that will make them sweat because I don’t want to put her into all the competitive stuff.” (Participant 12, female)
iv. Stigma and Bullying “It was hard for him [my son] to understand why the kids at school called him fat…so it’s kind of trying to help explain that stuff to him.” (Participant 6, male) “The problem that my son is facing is mental health…. He feels he is isolated because he is not able to relate to others and he has used the word ‘bullying’ quite often…. He hasn’t missed school or pretended to be sick because he doesn’t want to go, his friends still play with him, but he does not feel like he belongs.” (Participant 11, female) “There’s lot of blame and guilt that comes with your child who is overweight and you feel like everyone walking around and looks at you, [thinking] she’s not thin and no wonder her child’s not thin.” (Participant 1, female)
v. Geographical and Seasonal Issues “It’s tougher now too as we get into the winter.” (Participant 2, male) “Especially now when it’s dark at five, like shower, bed time, let’s go.” (Participant 11, female) “He’s had such a struggle always and we’ve been trying to reach out to get help, but it’s so hard. We live in a small community, about an hour away from here, but this was the closest that we could really get help.” (Participant 6, male)
**Time Constraints**
“Barring going into a restaurant, to find a quick option when you are like off to a game or something like that, it is impossible” (Participant 10, male) “I have been paying for the Y for six years … I only do the summer camps, because I finish work by 5:30–6 o’clock and there is no way I can get anywhere.” (Participant 11, female) “Getting your kids and family fed so you can get to the rest of the life…. You rush them through to get everything done, we are going back and forth from activity to home. You might have 10 or 15 min at home so you don’t have time to prep a great meal for your family…. Our lives are really busy; we’re gone every day of the week.” (Participant 9, female) “Sometimes you know you shouldn’t be doing this, but you have 5 min or 10 min to get supper ready and get to whatever it is you have to do. I work sometimes 14-h days.” (Participant 8, male) “She [my daughter] could play in the park for 9 or 10 h if you let her. It’s just, I’ve got her two days a week and if we go play at the park for 6–7 h then I’m going to get absolutely nothing done. So as much as I would love to let her just go, there are other things I have to do.” (Participant 8, male) “There’s so many things. Some days you get home from work and you had a plan, but there is no way, it’s just is not happening. For me it is all about dinners…I don’t meal prep, I don’t have time on weekends, I don’t do any of that so I get home and then I start making the dinners. Like [that’s] what really holds us back from doing anything.” (Participant 12, female) “A barrier to nutrition is again, time management.” (Participant 3, male)
**Parenting Issues**
i. Protecting Children’s Feelings “She [my daughter] is a very confident girl, and she sees herself as very pretty and she’s very popular. She’s got tons of friends, but I think the moment those words come out of my mouth, that she thinks I look at her differently…I feel like it could shatter her.” (Participant 12, female) “We didn’t talk a lot about the program to my daughter and I have … a bit of anxiety. I didn’t want to say why I sought it out. I didn’t want to say that… and whatever confidence they have, you don’t want to change that.” (Participant 4, female) “She has the most self-confidence … my biggest hope is that it stays…. She’s having fun and she’s enjoying it, and is keeping it positive. That is the part that I like, that I want to keep.” (Participant 7, female)
ii. Setting Appropriate Boundaries “It’s hard to say absolutely no screens ever, all day until Friday, until the weekend.” (Participant 5, female) “You don’t want it to feel like, ‘No, you can’t have this. No you cannot have this.’” (Participant 11, female) “It’s hard ‘cause you’re giving your kid withdrawals, like ‘No, you can’t do what everyone else is doing’” (Participant 3, male)
iii. Difficulties Associated with Relaying Program Content to Children “I go home and she doesn’t listen to me. She would listen to someone else standing in front of the room talking to everyone. She would be more likely to get something out of that, than me retelling what we learned because I’m Mom and I don’t know that much.” (Participant 4, female) “Parents don’t carry as much weight as the experts. Everything we took away from here was delivered by the parent. But if it’s not being delivered by Mom and Dad, it would be less [of a] chore.” (Participant 2, male)

**Table 6 ijerph-16-02171-t006:** Selected quotes related to parents’ recommendations for future paediatric overweight/obesity interventions.

**Greater Child Involvement**
i. Increased Accountability of Children “Take it one step further… if the kids were a little more involved along the way… something child-focused, so they can feel even more proud of their accomplishments and share amongst their peers too.” (Participant 1, female) “I think it would have been even more powerful to have … the child around the table with you to set those goals right because there’s a bit of a lack of commitment when they just don’t have that information coming right from the source.” (Participant 1, female) “I just think the kids were too excluded. Everything we took away from here was delivered by the parent… but if like the goal setting was done in a group environment…. I came up with the goals, we asked the kids, like what are some things we can do, but it’s just not the same right?” (Participant 2, male)
ii. Hands-on Activities for Children “There’s a lot of information sessions that were great for parents, but it would almost be great if you had one week information session, the next week you did a little lab session with the kids. Presented the information in that way, so you did information and then the practical.” (Participant 2, male) “We put [our] kids in the cooking class at the [grocery store]…. maybe that could’ve been a finale where the children got to make something and be proud of that accomplishment using key ingredients.” (Participant 9, female) “He [my son] wants a recap to see what we talked about and of course I do like a small summary just so that he is aware of what we talked about, but we also talked about the day that we had the chef came in… and he thought it would be a cooking class to teach him how to cook. He wanted that involvement, he wanted that extra piece…. He wanted to be a little more involved in the program.” (Participant 11, female)
iii. Peer Support and Interactions Among Children “I like the idea of exchanging numbers, not just for the parents, but for the children, like you guys make friends within the program.” (Participant 3, male) “If a peer group similar to this could be made for kids. The parents can get together and find useful solutions, while the kids play and make friends. They’d probably all go to different schools, but they’d have the exact same issues and they won’t judge.” (Participant 12, female) “Bring them together. We’re doing our thing, but they’re doing something on a different level…. That brings the kids together at their own language and pace, and builds some friendships too.” (Participant 9, female)
**Practical Information and Strategies**
i. Other Topics of Interest to Parents “You could have a time management type of component too.” (Participant 8, male) “Emotional intelligence so that we know what words to say.” (Participant 11, female) “Have a social worker come in and… connect with the schools and see what kind of support there is.” (Participant 11, female)
ii. Missing the ‘How’ “I’ve learned something… everything was great but I just felt like how to implement it.” (Participant 12, female) “We are more aware of what we need to do, but … we are missing that ‘how’.” (Participant 11, female) “I never got a clear idea as of how to talk to my kid and if there was somebody that came in that said, ‘When your kid reacts this way, you can say this, here’s a strategy to deal with certain answers.’ You know like salespeople they know, if you say, ‘Oh no, not right now’, then they have something ready.” (Participant 3, male)

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
