# Peer review of "Participants’ Perceptions of “C.H.A.M.P. Families”: A Parent-Focused Intervention Targeting Paediatric Overweight and Obesity"

_ijerph, 2019, doi:10.3390/ijerph16122171_

Round 1
Reviewer 1 Report
The paper is well written and informative in general. It was a qualitative research using focus groups to explore participants’ perceptions of “C.H.A.M.P. Families”. The study findings would be relevant to researchers in this field.
But the paper appeared to be very lengthy since only 2 focus groups with 12 parents were conducted. The readers would wonder if the information from the 2 focus groups would have reached the saturation point. In other words, the study might have missed some key findings due to a small number of participants. This was one of the main limitations of this study.
Another point was that the effectiveness of “C.H.A.M.P. Families” project in terms of prevention of childhood obesity was only based on a feasibility study ( reference 53) with a small sample size (n= 15 in year 1 and 25) using a pre-post study. This means that there was no robust study conducted in testing the effectiveness of “C.H.A.M.P. Families” project.
There was also a typo on reference 53.
Author Response
The paper is well written and informative in general. It was a qualitative research using focus groups to explore participants’ perceptions of “C.H.A.M.P. Families”. The study findings would be relevant to researchers in this field. But the paper appeared to be very lengthy since only 2 focus groups with 12 parents were conducted. The readers would wonder if the information from the 2 focus groups would have reached the saturation point. In other words, the study might have missed some key findings due to a small number of participants. This was one of the main limitations of this study.
Thank you for your time and consideration of this manuscript. We agree that the small sample size may have affected our ability to reach saturation. This issue, along with the lack of diversity among participants, prevents the generalizability of these findings. We have now acknowledged both of these limitations in the Discussion section (Lines 505-518).
Another point was that the effectiveness of “C.H.A.M.P. Families” project in terms of prevention of childhood obesity was only based on a feasibility study (reference 53) with a small sample size (n= 15 in year 1 and 25) using a pre-post study. This means that there was no robust study conducted in testing the effectiveness of “C.H.A.M.P. Families” project.
Thank you for this point. While C.H.A.M.P. Families was indeed an extension of the original C.H.A.M.P. project (and based upon some of the empirical findings from that program of research), C.H.A.M.P. Families was designed using a different model (i.e., parent-focused rather than child-focused as per the original C.H.A.M.P. program). The rationale for the parent-focused approach was based primarily on compelling evidence from the childhood obesity treatment intervention literature, and was developed with an evidence-informed theoretical framework. The rationale for this study is explained in the Introduction, with further details about the theoretical foundation and intervention design available in a previously published manuscript (Reilly et al., 2018; ref 74). Given that this is a pilot feasibility study, traditional methods of testing to measure effectiveness were not conducted; rather, the feasibility of the intervention is currently being evaluated using the RE-AIM (Reach, Effectiveness, Adoption, Implementation, Implementation, Maintenance) Framework (refs 56-59), which includes an effectiveness component. In short, the currently discussed C.H.A.M.P. Families program is novel and unique, and while an extension of the original C.H.A.M.P. program of research, it is sufficiently different to warrant its own evaluation (to which the current study makes an important contribution).
There was also a typo on reference 53.
Thank you for bringing this to our attention. The reference has now been edited (Lines 687-689).
Reviewer 2 Report
This study reviewed parents’ perspectives of C.H.A.M.P, but did not performed any scientific analysis. Although the number of participants is small (n=12 representing 7 children), it would be meaningful if the study involves any scientific data (e.g., BMI change of the children, change of daily calorie intake or physical activities…). If it is not possible to perform further analysis, I strongly recommend providing initial BMI of the children.
Furthermore, there is a possibility of selection bias in this study. The participants who attended actively with a good compliance might have better socioeconomic status and educational level than who did not participate in the focus group. This limitation has to be written in the discussion in detail.
Author Response
This study reviewed parents’ perspectives of C.H.A.M.P, but did not performed any scientific analysis. Although the number of participants is small (n=12 representing 7 children), it would be meaningful if the study involves any scientific data (e.g., BMI change of the children, change of daily calorie intake or physical activities…). If it is not possible to perform further analysis, I strongly recommend providing initial BMI of the children.
Thank you very much for this comment and suggestion. In addition to collecting qualitative data from focus groups, we gathered data on a number of outcomes to be analyzed quantitatively (e.g., children’s standardized body mass index [BMI-z], children’s physical activity measured via accelerometers, children’s health-related quality of life, etc.). While interesting, the presentation of these data is not in line with the purpose of the present study, which was to explore (qualitatively) the experiences of parents who participated in C.H.A.M.P. Families, as well parents’ perceived impact of the program on child and parental wellbeing (Lines 117-120). Findings related to these other important outcomes will be presented in separate publications. As for the reviewer’s suggestion to include the mean BMI-z of children, we agree that this is important information; in fact, the mean BMI-z of child participants at baseline was included in the Results section of the original submission (Line 212-213). Further, for consistency and because it may also be of interest to the readers, we have now included the mean BMI of parents at baseline as well (Lines 211-212).
Furthermore, there is a possibility of selection bias in this study. The participants who attended actively with a good compliance might have better socioeconomic status and educational level than who did not participate in the focus group. This limitation has to be written in the discussion in detail.
Thank you for this comment. This is an excellent point. We agree that the participants in our focus groups, and in the C.H.A.M.P. Families sample overall, were not particularly diverse in terms of their household income, educational attainment, and ethnicity. We have revised the manuscript to explicitly express this as a limitation (Line 506-517). With regard with to selection bias, we have provided some additional information in the limitations section (Lines 506-510) to better contextualize the individuals who did not participate in the focus groups.
Reviewer 3 Report
Congratulations to the authors. Qualitative analyses are necessary to direct strategies and interventions in risk groups. Childhood obesity must be understood as a complex public health problem.
Author Response
Comments and Suggestions for Authors:
Congratulations to the authors. Qualitative analyses are necessary to direct strategies and interventions in risk groups. Childhood obesity must be understood as a complex public health problem.
Response to Reviewers:
Thank you very much for your comments. We appreciate your time and consideration of our manuscript.
Reviewer 4 Report
The authors’ goal was to determine parents’ perceptions of their participation in a childhood obesity program (C.H.A.M.P.S. Families) and provide their suggestions for future childhood obesity programs. This assessment of parental perceptions is warranted due to their lack of examination within the context of a community-based, parent-only lifestyle intervention. This manuscript is a well-written, thorough assessment of parents’ perceptions.
Author Response
Comments and Suggestions for Authors:
The authors’ goal was to determine parents’ perceptions of their participation in a childhood obesity program (C.H.A.M.P.S. Families) and provide their suggestions for future childhood obesity programs. This assessment of parental perceptions is warranted due to their lack of examination within the context of a community-based, parent-only lifestyle intervention. This manuscript is a well-written, thorough assessment of parents’ perceptions.
Response to Reviewers:
Thank you very much for your comments. We appreciate your time and consideration of our manuscript.